# Quantum spin spiral ground state of the ferrimagnetic sawtooth chain

Roman Rausch[1][*], Matthias Peschke[2,3], Cassian Plorin[3],
Jürgen Schnack[4] and Christoph Karrasch[1]

**1** Technische Universität Braunschweig, Institut für Mathematische Physik,
Mendelssohnstraße 3, 38106 Braunschweig, Germany
**2** Institute for Theoretical Physics Amsterdam and Delta Institute for Theoretical Physics,
University of Amsterdam, Science Park 904, 1098 XH Amsterdam, The Netherlands
**3** Department of Physics, University of Hamburg,
Notkestraße 9, 22607 Hamburg, Germany
**4** Faculty of Physics, Bielefeld University, Universitätsstr. 25, 33615 Bielefeld, Germany

⋆ r.rausch@tu-braunschweig.de

## Abstract

The ferrimagnetic phase of the sawtooth chain with mixed ferromagnetic nearest-neighbour interactions $J$ and antiferromagnetic next-nearest-neighbour interactions $J'$ (within the isotropic Heisenberg model) was previously characterized as a phase with commensurate order. In this paper, we demonstrate that the system in fact exhibits an incommensurate quantum spin spiral. Even though the ground state is translationally invariant in terms of the local spin expectations $\langle S_i \rangle$, the spiral can be detected via the connected spin-spin correlations $\langle S_i \cdot S_j \rangle - \langle S_i \rangle \cdot \langle S_j \rangle$ between the apical spins. It has a long wavelength that grows with $J'$ and that soon exceeds finite-system sizes typically employed in numerical simulations. A faithful treatment thus requires the use of state-of-the-art simulations for large, periodic systems. In this work, we are able to accurately treat up to $L = 400$ sites (200 unit cells) with periodic boundary conditions using the density-matrix renormaliztion group (DMRG). Exploiting the SU(2) symmetry allows us to directly compute the lowest-energy state for a given total spin. Our results are corroborated by variational uniform matrix product state (VUMPS) calculations, which work directly in the thermodynamic limit at the cost of a lower accuracy.



# 1 Introduction

## 1.1 Frustration and ferrimagnets

The Lieb-Mattis theorem [1, 2] states that for a bipartite Heisenberg system with antiferromagnetically coupled sublattices A and B, the ground state has the total spin $\left|S_{\mathrm{max},A} - S_{\mathrm{max},B}\right|$, where $S_{\mathrm{max},A}$ and $S_{\mathrm{max},B}$ are the maximally possible spins of the respective sublattices. For the common case that the sublattices are equivalent (i.e., consist of atoms with the same spin) and are of equal size, this yields a singlet ground state. If they are inequivalent, one obtains a ferrimagnet with a predictable ground-state spin and opposite orientations of the sublattice polarizations.

The situation gets more complicated if frustration is allowed to enter into the picture and the couplings become non-bipartite. In addition, mixed ferro- and antiferromagnetic couplings can result in a ferrimagnet even for equivalent sublattices. This is the case we will consider in this paper.

Interacting localized spins are commonly described by the Heisenberg model, which can be generally written down as

$$H = \sum_{i<j} J_{ij} \mathbf{S}_i \cdot \mathbf{S}_j \,, \tag{1}$$

where $\mathbf{S}_i = (S_i^x, S_i^y, S_i^z)$ is the spin operator at site $i$ and $J_{ij}$ are the coupling constants that define the geometry. Since the Heisenberg Hamiltonian commutes with each component of the vector of the total spin $\mathbf{S}_{\mathrm{tot}} = \sum_i \mathbf{S}_i$ as well as with its square,

$$[H, \mathbf{S}_{\mathrm{tot}}^2] = 0 \,, \tag{2}$$

there exists a simultaneous eigenbasis of $H$ and $\mathbf{S}_{\mathrm{tot}}^2$, and the ground state can in principle take any value of $S_{\mathrm{tot}}$ between 0 and $LS$, where $L$ is the number of sites and where $S_{\mathrm{tot}}$ is determined from

$$\left\langle \mathbf{S}_{\mathrm{tot}}^2 \right\rangle = S_{\mathrm{tot}}(S_{\mathrm{tot}} + 1) \,. \tag{3}$$

Intuitively, it is clear that for the mixed-coupling case, where $J_{ij} > 0$ for some sites and $J_{ij} < 0$ for others, there should be a region where neither the singlet state $S_{tot}/L = 0$, nor the ferromagnet $S_{tot}/L = S$ minimizes the energy and the ground state will have some partial polarization $0 < S_{tot}/L < S$. For later purposes, we also introduce the quantum number $M_{tot}$ related to the conservation of the $z$-component, i.e., a U(1) spin symmetry:

$$\left\langle S_{tot}^z \right\rangle = \sum_i \left\langle S_i^z \right\rangle = M_{tot} \,. \tag{4}$$

Since the Lieb-Mattis theorem does not hold anymore in the frustrated case, $S_{tot}$ must be determined from a full many-body calculation. In addition, frustration may favour non-collinear spin-spiral or canted states [3–5]. This should be understood in a quantum sense: A finite polarization can be interpreted as a spontaneous breaking of the SU(2) symmetry down to U(1), with the total spin pointing along the quantization axis. Hence, there is no classical non-collinear order, where the angle of the classical vector $\langle \mathbf{S}_i \rangle = (\langle S_i^x \rangle, \langle S_i^y \rangle, \langle S_i^z \rangle)$ would vary as a function of the site index $i$ [6]. However, the spin-spin correlations may peak at a value of the wavevector not equal to 0 or $\pi$, which signals non-collinearity. An alternative diagnostic is the susceptibility to small twists [3, 7].

From an experimental perspective, there are several examples of systems with mixed ferro- and antiferromagnetic interactions. Some one-dimensional cuprates can be described as extended $S = 1/2$ Heisenberg chains with nearest-neighbour exchange $J < 0$ and next-nearest-neighbour exchange $J' > 0$ [8–11]. Another case are single-molecule magnets (SMM), a subclass of which is based on Mn ions of various sizes and geometries [12]. The largest to date are the {Mn$_{70}$} and {Mn$_{84}$} wheels [13, 14] with $S = 2$ Mn(III) centres and a surprisingly low total spin of $S_{tot} = 5 - 8$. These are finite, but still quite challenging systems, which have received thorough theoretical attention only recently [15], pointing to a necessity of mixed FM-AFM interactions to achieve such a low spin.

## 1.2 The sawtooth chain

In this work, we focus on another FM-AFM system, the "sawtooth" or "delta" ($\Delta$) chain [3, 5, 16–21], which consists out of vertex-sharing triangles and which is probably the simplest 1D geometry with geometrical frustration[1] (see Fig. 1). It features a two-site unit cell with alternating "apical" (A) and "basal" (B) spins. The corresponding Heisenberg Hamiltonian is given by

$$H = J \sum_i \left( \mathbf{S}_i^A \cdot \mathbf{S}_i^B + \mathbf{S}_i^A \cdot \mathbf{S}_{i+1}^B \right) + J' \sum_i \mathbf{S}_i^B \cdot \mathbf{S}_{i+1}^B \,, \tag{5}$$

where the sums run over the unit cells ($L = 2N_{cells}$). $J$ and $J'$ are the exchange coupling constants, one of which sets the energy scale. The sawtooth chain comes essentially in two variants: An AFM-AFM one with both $J > 0$ and $J' > 0$ [3, 21–28]; and a mixed FM-AFM one with $J < 0$ and $J' > 0$ [5, 17–20].

Experimentally, the sawtooth geometry is found for atacamite (AFM-AFM, $S = 1/2$, $J'/J \approx 3.29$) [29], for the ring molecule Fe$_{10}$Gd$_{10}$ [30] (FM-AFM, mixed $S = 5/2$ and $S = 7/2$, $J'/|J| \approx 0.65$) and for a malonato-bridged Cu complex [17] (FM-AFM, $S = 1/2$, $J'/|J| \approx 0.91$).

In this paper, we investigate the homogeneous FM-AFM $S = 1/2$ case, relevant for the last material. We note that $J'/|J| \approx 0.91$ [17] is within the interesting region $J'/|J| \sim 1$, where the couplings are of equal strength. We will thus pay special attention to the point $J'/|J| = 1$ in this work.

---

[1]While the kagome lattice is also composed out of vertex-sharing triangles, it is complicated by closed loops. The sawtooth chain, on the other hand, is a special case of a delta tree without closed loops [16].

If the coupling between the apical spins cannot be neglected, one needs to add the corresponding term to the Hamiltonian:

$$H_\gamma = \gamma J' \sum_i \mathbf{S}_i^A \cdot \mathbf{S}_{i+1}^A.$$

(6)

For $\gamma = 1$, the Hamiltonian reduces to an extended Heisenberg chain [31,32]. In this work, we only deal with the sawtooth limit $\gamma = 0$.

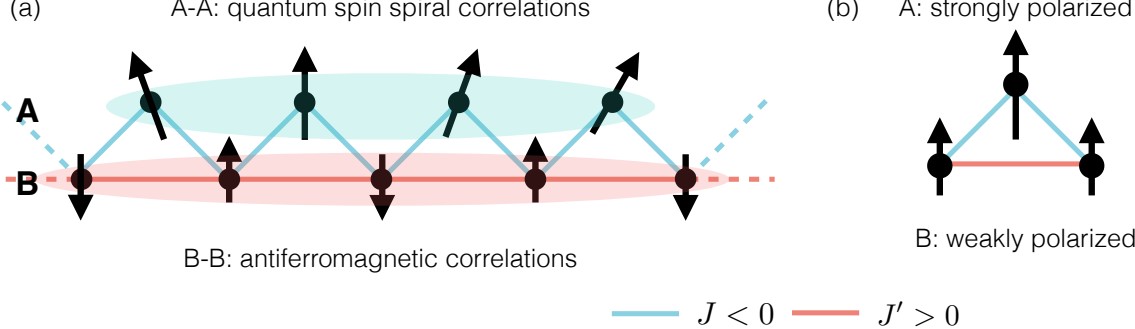

Figure 1: Sketch of the FM-AFM sawtooth chain and the proposed magnetic order in the ferrimagnetic phase. The apical and basal sites are labeled as "A" and "B", respectively. (a) Schematic visualization of the spin-spin correlations. (b) Visualization of the spin polarization pattern. Both the quantum spin spiral (A-A) and the AFM order (B-B) can only be detected via the connected spin-spin correlations and not via the polarization.

## 1.3 Previous results

We briefly summarize the state of knowledge regarding the $S = 1/2$ AFM-AFM sawtooth chain. It features three phases as a function of $J'/J$, namely gapless antiferromagnetic, gapped dimerized, and gapless non-collinear [3,21]. The non-collinear phase has not been much explored to the best of our knowledge; most studies have focused on the dimerized phase, where a valence-bond solid (VBS) ground state appears for $J' = J$, which has solitonic excitations [21–23]. Flat magnon bands appear at the specific point $J'/J = 1/2$ [28,33–35] and lead to an exceptionally large jump from full saturation to half saturation due to localized magnons [24–27].

We will now recapitulate prior results for the mixed-coupling (FM-AFM) $S = 1/2$ sawtooth chain, which is the subject of this paper. A first theoretical treatment used exact diagonalization (ED) as well as density-matrix renormalization group (DMRG) calculations for odd chain lengths $L = 7, 11, 15, \ldots, 31, 47, 67$ [18]. For $J' = 0$ the system must clearly be ferromagnetic (with $S_{\text{tot}}/L = 1/2$), and it is found that ferromagnetism persists for small $J'/|J|$. A transition to a ferrimagnetic phase is observed for $J'/|J| = 0.5$. The total spin per site was found to follow $S_{\text{tot}}/L = (L-1)/(4L) + 1/2L$ and thus approaches $1/4$ for $L \to \infty$. No further statements on the nature of the ground state were given in this paper [18].

Later works mostly dealt with particular regimes and specific questions such as the crossover between the Ising and Heisenberg limits [19], the comparison of magnetization curves with the experiment [20], or the thermodynamics around the critical point [36], in particular with an application to the ferromagnetic molecule $Fe_{10}Gd_{10}$ [37].

Recently, the system received renewed interest, and the properties of the ferrimagnetic phase (including the finite-$\gamma$ case) were investigated in great detail using the DMRG for finite systems with open and periodic boundary conditions (exploiting the U(1) spin symmetry and

using bond dimensions up to $\chi = 8000$) [5]. For $\gamma = 0$, previous results [18] were confirmed, whereby the total spin per site for even rings up to $L \sim 50$ is of the form $S_{\text{tot}}/L = 1/4 + 1/L$, implying that $S_{\text{tot}}/L$ can be extrapolated in the thermodynamic limit. The phase was characterized as a commensurate ferrimagnet that only becomes incommensurate for $\gamma > 0$.

In this paper, we revisit the sawtooth chain using DMRG with periodic boundary conditions. We exploit the full SU(2) spin symmetry of the problem, access rings larger by about a factor of 10 and come to a different conclusion about the nature of the ground state: We find that the total spin takes irrational values $0.25 \lesssim S_{\text{tot}}/L \lesssim 0.28$ and probably reaches $1/4$ only for $J' \to \infty$. The ground state is characterized by an incommensurate quantum spin spiral in the connected apex-apex correlations $\left\langle \mathbf{S}_i^A \cdot \mathbf{S}_j^A \right\rangle - \left\langle \mathbf{S}_i^A \right\rangle \cdot \left\langle \mathbf{S}_j^A \right\rangle$. The main reason for this discrepancy is that the wavelength of the spiral is generally very large; it grows with $J'$ and soon exceeds finite-system sizes that were considered in prior works. Another confounding factor are the very small energy gaps between the various spin sectors (in particular for large values of $J'/|J|$), which renders the exploitation of the SU(2) symmetry extremely beneficial for this problem.

## 2 Technical details

### 2.1 Finite systems

For finite systems, we use the DMRG algorithm, which is a well-established approach to compute ground state properties of 1D problems variationally in the space of matrix-product states [38, 39]. Its effectiveness rests on the so-called "area law" for the entanglement entropy [40], which guarantees a low entanglement for ground states of short-ranged Hamiltonians on 1D chains with open boundary conditions, which can be used to truncate the full Hilbert space to a much smaller relevant space. The main control parameter of this truncation is called the "bond dimension" $\chi$. We use the one-site algorithm with a subspace expansion method [41] to dynamically increase the bond dimension during the iterations and have selectively checked that the two-site algorithm [39] yields the same results.

It was shown that for the sawtooth chain, the interpretation of results obtained for open boundary conditions can be subtle and complicated [5], probably because the Friedel oscillations at the open ends interfere with the delicate spin order and the small energy gaps. Hence, it is better to use periodic boundary conditions, but this generally diminishes the effectiveness of the DMRG and one needs to employ extremely large bond dimensions. However, we can counteract this by exploiting the SU(2) symmetry of the problem [42, 43] and access very large effective bond dimensions $\chi_{\text{eff}}$, while numerically working with a much smaller and tractable $\chi_{\text{SU(2)}} \ll \chi_{\text{eff}}$.

The exploitation of the SU(2) symmetry boils down to using the Wigner-Eckart theorem which states that under SU(2) symmetry, matrix elements only depend on the spin projections via Clebsch-Gordan coefficients that can be separated out. This means that the local blocks within the DMRG ansatz state effectively correspond to $2S_{\text{block}} + 1$ states for every intermediate value of $S_{\text{block}}$. The gain is diminished for high polarizations, as one is typically only interested in the sector with the maximal spin projection $M_{\text{tot}} = S_{\text{tot}}$, which can also be efficiently obtained with a U(1) code. Nevertheless, SU(2) remains beneficial, as it exactly projects out unwanted total-spin states with the same $M_{\text{tot}}$, allows us to compute the lowest energy in every sector of the total spin $E_0(S_{\text{tot}})$ and to distinguish between a low-spin and a high-spin solution. Table 1 shows the typical bond dimensions used in this work. Large values of $S_{\text{tot}}$ require a stable computation of Wigner $3j$ and $6j$ symbols for large inputs, for which we use the WIGXJPF

Table 1: Typical bond dimensions and system sizes used in this work in terms of the system size $L$ and the symmetries exploited in the algorithm. For finite systems, we always use periodic boundary conditions. For $L = \infty$ and the case of SU(2), we can only access the sector $S_{\text{tot}} = 0$. Note that for SU(2) and the $S_{\text{tot}} = 0$ sector, the effective bond dimension is $\chi_{\text{eff}} = g \cdot \chi_{\text{SU(2)}}$ with a gain factor of $g \sim 5 - 10$.

| $L$ | symmetry | $\chi$ or $\chi_{\text{SU(2)}}$ |
|---|---|---|
| 40 | SU(2) | 500 |
| 60 | SU(2) | 2000 |
| 100 | SU(2) | 2000 |
| 200 | SU(2) | 2000 |
| 252 | SU(2) | 3000 |
| 300 | SU(2) | 4000 |
| 400 | SU(2) | 6500 |
| $\infty$ | no symm. | 1000-1200 |
| $\infty$ | U(1) | 3000-4000 |
| $\infty$ | SU(2) | 3000 |

library [44].

For finite systems, we identify the absolute ground state from the minimum of $E_0(S_{\text{tot}})$. The error is assessed by computing the variance per site

$$\text{Var}(E)/L = \left(\langle H^2 \rangle - \langle H \rangle^2\right)/L. \tag{7}$$

As we show in App. A.1, this measure is proportional to the actual error in the ground-state energy density, allowing us to put error bars on the computed energies. We choose the bond dimension such that $\text{Var}(E)/L \leq \mathcal{O}\left(10^{-6}\right)$ around the minimum of $E_0(S_{\text{tot}})$.

In addition, we can assess the accuracy by comparing with results of Lanczos diagonalization for smaller system sizes up to $L = 36$ (see App. A.1). In this case, we exploit the U(1) symmetry and the conservation of the total momentum and extract the multiplets of the total spin from the degeneracies in the spectrum.

Since the variance per site has the dimension of energy squared and its scale changes with $J$ and $J'$, we ensure that the largest parameter in the Hamiltonian is of modulus 1: First, we set $J = -1$ and increase $J'$ up to $J' = 1$. Then, we keep $J' = 1$ and let $J$ go to zero. Only the ratio $J'/|J|$ matters for the phase diagram.

The main advantage of working with a finite system is the high accuracy of the DMRG with the SU(2) symmetry. The main disadvantage are the finite-size effects which become quite severe for the given problem, even for system sizes of $\mathcal{O}(10^2)$ sites, as will be shown below.

## 2.2 Infinite systems

For infinite boundary conditions, we use the variational uniform matrix-product state (VUMPS) formalism [45, 46], which is based on the time-dependent variational principle and offers improved efficiency over the original infinite DMRG [38]. Our numerical unit cell encompasses two physical unit cells (4 sites) in order to allow for AFM order. While finite-size effects are eliminated, this approach comes at the disadvantage that the exploitation of symmetries is limited: We can only use $S_{\text{tot}} = 0$ in the case of SU(2) and only a rational $M_{\text{tot}}$ within the unit

cell in the case of U(1). Since the finite-system data indicates that the physical ground state is generally not a spin singlet, we can thus not employ the highly efficient SU(2) numerics.

If U(1) symmetries are exploited in the infinite system, there is to the best of our knowledge no practical way to compute the value of $S_{\text{tot}}/L$ in the $M_{\text{tot}} = 0$ sector. In order to access $S_{\text{tot}}/L$, we switch off the symmetry altogether [47]. Within the degenerate set of $2S_{\text{tot}} + 1$ states, the DMRG tends to converge to the $M_{\text{tot}} = S_{\text{tot}}$ sector, whose entanglement is minimal. In this case, $\left\langle S_i^{z,\alpha} \right\rangle$ and $\left\langle S_i^{x,\alpha} \right\rangle$ ($\alpha = A, B$) take finite, translationally invariant values ($\left\langle S_i^{y,\alpha} \right\rangle$ vanishes by time-reversal symmetry), and make up the dominant contribution to the total spin formula in Eq. (3) [48]:

$$S_{\text{tot}}^{\alpha}/N_{\text{cells}} \approx \sqrt{\left\langle S_i^{x,\alpha} \right\rangle^2 + \left\langle S_i^{z,\alpha} \right\rangle^2}, \quad S_{\text{tot}}/L \approx \left( S_{\text{tot}}^A/N_{\text{cells}} + S_{\text{tot}}^B/N_{\text{cells}} \right)/2. \tag{8}$$

As an error estimate for infinite systems, we look at the convergence with respect to the bond dimension $\chi$ (see App. A.3).

## 2.3 Expectation values, spin structure factor

A polarized ground state has a $(2S_{\text{tot}} + 1)$-fold degeneracy, and one needs to specify w.r.t. to which of these states expectation values are computed. Within the SU(2)-symmetric approach for the finite system, we can directly access each member of the multiplet [42, 43]. In the infinite system, one can straightforwardly only determine the state with $M_{\text{tot}} = 0$ using the U(1)-symmetric algorithm; or the state with $M_{\text{tot}} = S_{\text{tot}}$ if no symmetries are exploited.

In order to demonstrate the existence of a quantum spin spiral, we want to compute the static spin structure factor, i.e., the Fourier transform of the spin-spin correlations. For a local operator $O_i^{\alpha}$, we define the connected correlation function as

$$\left\langle O_j^{\alpha} O_l^{\beta} \right\rangle_c = \left\langle O_j^{\alpha} O_l^{\beta} \right\rangle - \left\langle O_j^{\alpha} \right\rangle \left\langle O_l^{\beta} \right\rangle. \tag{9}$$

For the specific case of a ring with an even, finite number of unit cells $N_{\text{cells}}$, the static spin structure factor is obtained as follows:

$$\begin{aligned} C^{\alpha\beta}[O](k) &= \frac{1}{N_{\text{cells}}} \sum_{j,l=-N_{\text{cells}}/2+1}^{N_{\text{cells}}/2} e^{ik(j-l)} \left\langle O_j^{\alpha} O_l^{\beta} \right\rangle_c \\ &= \sum_{l=-N_{\text{cells}}/2+1}^{N_{\text{cells}}/2} e^{ik(j_0-l)} \left\langle O_{j_0}^{\alpha} O_l^{\beta} \right\rangle_c \\ &= \left\langle O_{j_0}^{\alpha} O_{j_0}^{\beta} \right\rangle_c + e^{ikN_{\text{cells}}/2} \left\langle O_{j_0}^{\alpha} O_{j_0+N_{\text{cells}}/2}^{\beta} \right\rangle_c \\ &\quad + \sum_{d=1}^{N_{\text{cells}}/2-1} \left[ e^{ikd} \left\langle O_{j_0}^{\alpha} O_{j_0+d}^{\beta} \right\rangle_c + e^{-ikd} \left\langle O_{j_0}^{\alpha} O_{j_0-d}^{\beta} \right\rangle_c \right]. \end{aligned} \tag{10}$$

We have assumed translational invariance, so that the result is independent of the site $j_0$, and have rewritten the summations in terms of the distance $d$. In the infinite system, one can evaluate the same equation for $N_{\text{cells}} \to \infty$, which on a technical level is achieved by using the MPS transfer matrix [46]. For $\alpha = \beta$, real matrix elements, and neglecting the second term that extends across the whole system, Eq. (10) reduces to a cosine transform:

$$C^{\alpha\alpha}[O](k) \approx \left\langle O_{j_0}^{\alpha} O_{j_0}^{\alpha} \right\rangle_c + 2 \sum_{d=1}^{N_{\text{cells}}/2-1} \cos(kd) \left\langle O_{j_0}^{\alpha} O_{j_0+d}^{\alpha} \right\rangle_c. \tag{11}$$

In the finite case with SU(2) symmetry being exploited, we compute[2]

$$C^{\alpha\beta}[\mathbf{S}] = C^{\alpha\beta}[S^x] + C^{\alpha\beta}[S^y] + C^{\alpha\beta}[S^z], \tag{12}$$

and $k$ can only take discrete values $k = 2\pi n/N_{\text{cells}}$ with $n = 0, 1, \dots N_{\text{cells}} - 1$. Due to the SU(2) symmetry of the problem, the first term in Eq. (9) is independent of $M_{\text{tot}}$ for the vector-vector correlations. In order to subtract the correct asymptotic value (and avoid a divergence at $k = 0$), the second term is evaluated for $M_{\text{tot}} = S_{\text{tot}}$. In the infinite case with U(1) symmetry being exploited, we compute

$$\lim_{N_{\text{cells}} \to \infty} C^{\alpha\beta}[S^z], \tag{13}$$

where $k$ can take continuous values. In this case, both terms in Eq. (9) depend on the choice of $M_{\text{tot}}$, while we can only access the sector $M_{\text{tot}} = 0$. However, we observe that the first term in Eq. (9) does not take a finite asymptotic value and that the second term vanishes (see Sec. A.4 and Fig. 16).

While the two correlation functions in Eqs. (12) and (13) do not coincide exactly, they can both be used to demonstrate the existence of a spin spiral and to determine its wavevector $k_{\text{peak}} \neq 0, \pi$.

## 3 Exemplary case $J = -1$, $J' = 1$

Figure 2 shows the energy profile $E_0(S_{\text{tot}})$ for finite systems (compared with the infinite one) in the interesting case of $J = -1$, $J' = 1$. For finite systems, the total spin per site must be rational, and we find the lowest-energy state (i.e., the true ground state) in the sector $S_{\text{tot}}/L = 27/100$ for $L = 100, 200$ and in the sector $S_{\text{tot}}/L = 68/252$ for $L = 252$. In all cases, these are the closest possible rational values to 0.270, so that this result seems converged w.r.t. the system size. The infinite-system result (without symmetries) appears to be irrational, and to the leading digits we obtain $S_{\text{tot}}/L \approx 0.270(2)$. Thus, our results are not in agreement with $S_{\text{tot}}/L = 1/4$ that was obtained before [5, 18], motivating a deeper investigation.

The ground-state energies are in good agreement between the finite- and infinite-system calculations (both for the case that no symmetry and that U(1) symmetry is exploited). In the curve $E_0(S_{\text{tot}})/L$, we observe a steep barrier towards high spins and a much shallower barrier towards small spins, where the energy density $E_0/L$ varies only in the fifth digit after the decimal point for the system sizes considered. Moreover, the energy gaps become smaller with larger system sizes, indicating gapless excitations, but only with respect to a decrease of the total spin.

It is notable that large, macroscopic changes in the total spin have very small gaps. Even the sector $S_{\text{tot}} = 0$ is very close in energy to the ground state, as was noticed before [5]; the inset of Fig. 2 shows that the singlet energy approaches the ground-state energy for large bond dimensions. This is, however, not an effect of frustration and is already observed for the FM Heisenberg chain.[3]

---

[2]Note that within the SU(2)-symmetric approach, the question of accessing individual $x, y, z$-components of either $\left\langle \mathbf{S}_j^\alpha \cdot \mathbf{S}_l^\beta \right\rangle$ or $\left\langle \mathbf{S}_j^\alpha \right\rangle$ becomes meaningless. Technical details can be found in Refs. [42, 43].

[3]For the FM Heisenberg chain, the total spin needs no calculation, as one can analytically show that it is maximal [49].

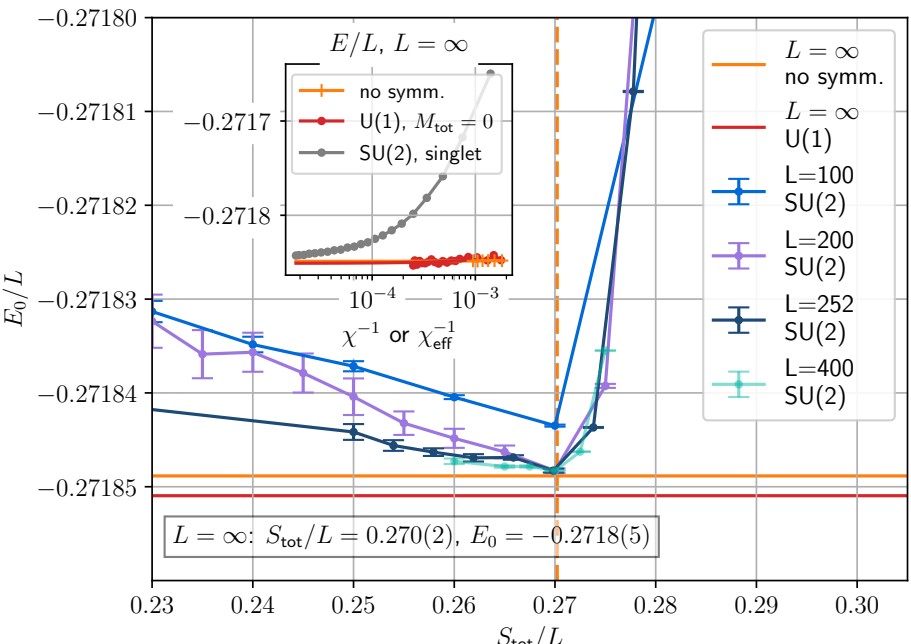

Figure 2: Lowest energies of the sawtooth chain with $J = -1$, $J' = 1$ in various sectors of the total spin $S_{\text{tot}}$ for finite systems, obtained using the SU(2)-symmetric DMRG with periodic boundary conditions. The lowest energy among the sectors is compared with the infinite-system (VUMPS) calculations, where one cannot target a specific sector of $S_{\text{tot}}$ (horizontal lines). The vertical dotted line shows $S_{\text{tot}}/L$ computed according to Eq. 8 for the infinite system without symmetries. The respective bond dimensions can be found in Tab. 1, and the estimation of error bars is outlined in App. A.1. The inset shows the energy of the infinite system as a function of the inverse (effective) bond dimension $\chi_{(\text{eff})}^{-1}$ (solid lines are inter/extrapolations). The two-site variance for the infinite system [45] is of the order of $10^{-5}$ (no symmetries), $10^{-4}$ [U(1)] and $10^{-3} \sim 10^{-4}$ [SU(2)].

## 4 Total spin in the ferrimagnetic phase

We now study the behaviour when moving away from the point $J = -1$, $J' = 1$. Figure 3 shows $S_{\text{tot}}/L$ as a function of $J'/|J|$ for different system sizes. The transition from the ferromagnet $S_{\text{tot}}/L = 1/2$ to the ferrimagnet at $J'/|J| = 0.5$ is unambiguous.

At the quantum critical point $J'/|J| = 0.5$, the DMRG finds that all values of the total spin are degenerate within the numerical accuracy, whereas Lanczos (as well as full SU(2) diagonalization for smaller systems) indicates that only the values $S_{\text{tot}} = L/2, L/2-1, \ldots, L/4, 0$ are degenerate.

In the ferrimagnetic phase, we observe the following features: (a) After crossing the quantum critical point, $S_{\text{tot}}/L$ jumps discontinuously to a value slightly above (but different from) 0.25. (b) One can reach convergence for $S_{\text{tot}}/L$ w.r.t. the accessible system size up to $J'/|J| \lesssim 1$. (c) For $J'/|J| \gtrsim 1$ it appears that we cannot access systems that are large enough to obtain convergence, and $S_{\text{tot}}/L$ features plateaus at various values of the spin. In particular, $S_{\text{tot}}/L$ at some point jumps to a low-spin state $0.005 \sim 0.01$ and eventually to zero. (d) The value of $J'/|J|$ where this jump happens increases with the system size.

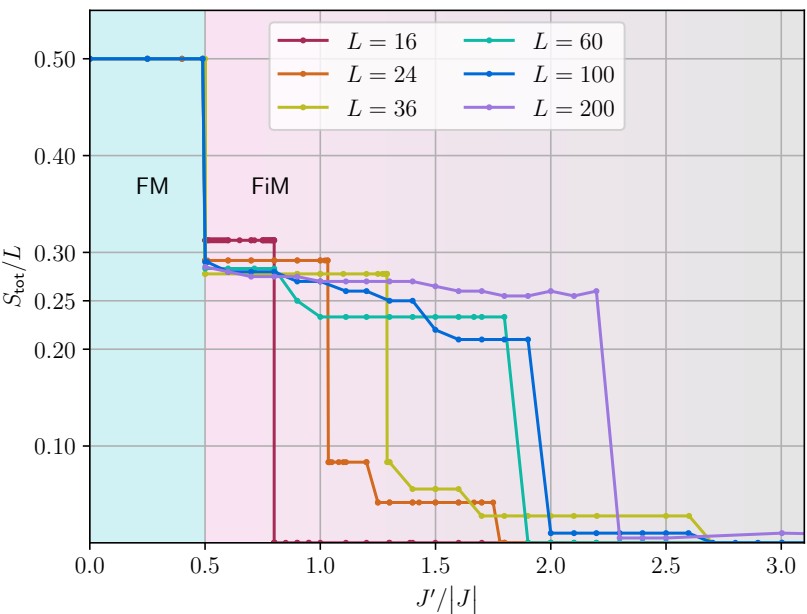

Figure 3: Total ground-state spin per site $S_{\text{tot}}/L$ for various finite systems with periodic boundary conditions. For $L \leq 36$, the results were obtained using exact diagonalization (Lanczos algorithm). For the other values, we used the DMRG with SU(2) symmetries and identified the true ground state from the minimum of $E_0(S_{\text{tot}})$ similarly to Fig. 2. FM and FiM denote the ferromagnetic and ferrimagnetic phase, respectively.

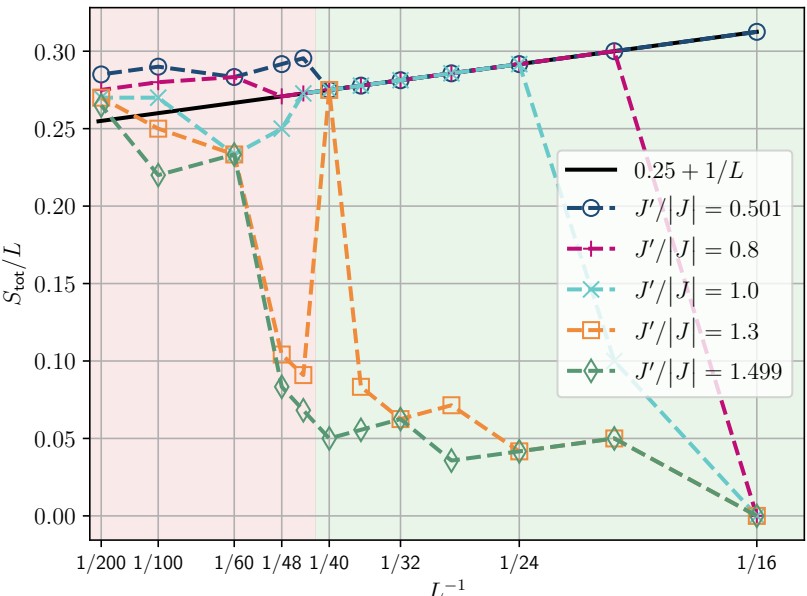

Figure 4: The same as Fig. 3 but plotted as a function of the inverse system size $L^{-1}$ for various values of $J'/|J|$. The curve $f(L^{-1}) = 0.25 + L^{-1}$ is shown for comparison. For small $J'/|J|$, the points collapse to $f(L^{-1})$ for a certain range of $L$ (green area), but this is not the case for larger $L$ or larger $J'/|J|$ (red area).

The chaotic behaviour with respect to the system size already points towards incommensu-

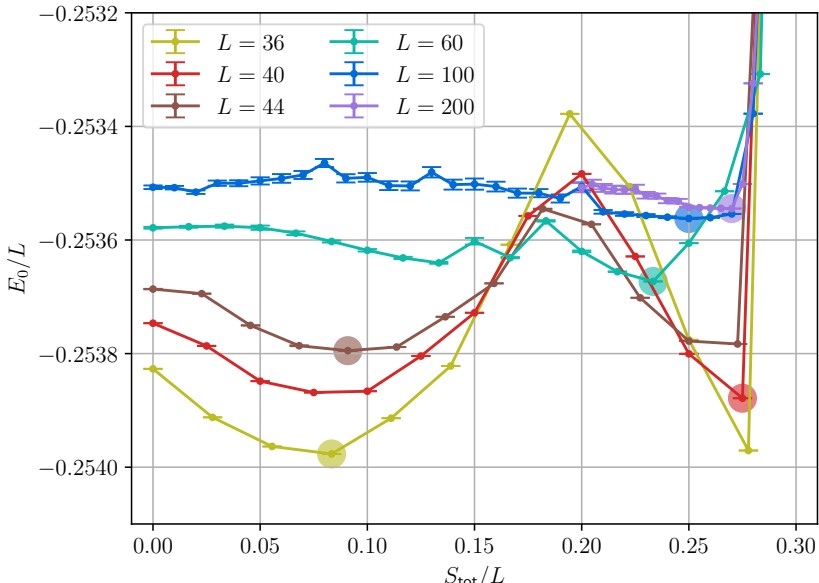

Figure 5: The same as in Fig. 2, but for $J'/|J| \approx 1.3$ and different system sizes $L$. The circle marks the absolute minimum. It is evident that small systems waver between two minima before eventually settling in the high-spin minimum, cf. the orange line in Fig. 4.

rate behaviour, while the observation (d) suggests that the low-spin state is a finite-size effect. We investigate these questions in more detail below.

In order to shed light on the discrepancy between our data and prior results, we now plot the same data as a function of $1/L$ in an attempt to perform an extrapolation w.r.t. the system size (see Figs. 4 and 5), as was done in Refs. [5,18]. One observes that for not too large $J'/|J|$, the points start collapsing on the curve $S_{tot}/L = 1/4 + 1/L$, which makes it tempting to extrapolate $S_{tot}/L = 1/4$ in the thermodynamic limit. However, if we increase the system size further instead of extrapolating, we find that the behaviour becomes chaotic somewhere around $L = 44$. For $J'/|J| \gtrsim 1.5$, the results in fact do not fall on $S_{tot}/L = 0.25 + 1/L$ at all. In both cases, this indicates that an intrinsic length scale is surpassed, and this length scale increases with $J'/|J|$. It is thus essential to access systems larger than this length scale (green area of Fig. 4). This is probably a reason for the discrepancy between our results and previous works.

Next, we compare the behaviour of large, finite systems $L = 100, 200$ with results obtained directly in the thermodynamic limit via the VUMPS algorithm (see Fig. 6). One finds that the infinite-system result for $S_{tot}/L$ is in good agreement with the one for $L = 100$ up to $J'/|J| \approx 1$ and with the one for $L = 200$ up to $J'/|J| \approx 1.5$ (and still in moderate agreement up to $J'/|J| \approx 2.2$). The crosses in the plot indicate energetically close sectors of the total spin for the finite system, illustrating again that the energy minimum is extremely shallow. Finite-size wavering within these minima is thus not surprising (cf. Fig. 5). One also notices that a second shallow minimum ($S_{tot}/L \sim 0.01$) develops around $J'/|J| \gtrsim 2$ at very low spins and eventually becomes the absolute one. This jump to low spins is not found in the infinite system. Thus, we concur with Ref. [5] that this jump is most likely a finite-size effect.

One should note that for $J'/|J| = \infty$ (equivalent to $J = 0, J' = 1$), the apical spins become free spins. Thus, the ground state is degenerate for all values of $S_{tot}/L = 0 \ldots 1/4$ [18]. This raises the question what the effect of an infinitesimal $J < 0$ is. Our numerical methods are ill-posed to answer this question because any gap will also be infinitesimal. Nevertheless,

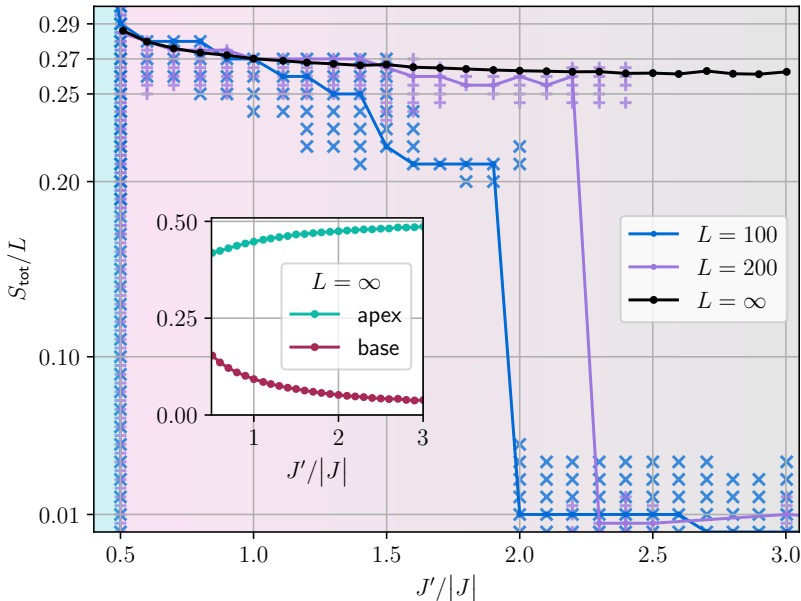

Figure 6: The same as Fig. 3, but including DMRG data obtained for the infinite system (VUMPS) without exploiting symmetries. In addition, the crosses indicate sectors of $S_{tot}$ in the finite system for which the energy density is within $10^{-3}$ of the ground-state energy density, illustrating the extreme shallowness of the energy minima (cf. Fig. 7). The inset shows the (translationally invariant) apical and basal polarization $\sqrt{\left\langle S_i^{x,\alpha}\right\rangle^2 + \left\langle S_i^{z,\alpha}\right\rangle^2}$ ($\alpha = A, B$) for $L = \infty$.

we note the following: The inset of Fig. 6 displays the polarization of the apical and basal spins for the infinite system computed without symmetries, i.e., in the sector $M_{tot} = S_{tot}$. The polarization is translationally invariant, i.e., $\left\langle S_i^{\alpha,x}\right\rangle$ and $\left\langle S_i^{\alpha,z}\right\rangle$ are independent of $i$ within numerical inaccuracies (in the sector $M_{tot} = 0$, one finds $\left\langle S_i^{z,A}\right\rangle \approx \left\langle S_i^{z,B}\right\rangle \approx 0$), which we can also confirm for finite systems. We see that the apical spins get more and more polarized towards $1/2$, while the basal spins approach 0. This suggests that in the limit $J'/|J| \to \infty$, one approaches $S_{tot}/L = 1/4$ very slowly. However, since the infinite-system calculations become progressively more difficult, we are only able to reliably treat $J'/|J| = 3 \sim 4$ and cannot exclude the possibility that additional effects happen for larger values.

Figure 7 is a schematic summary of the findings discussed in this chapter.

# 5 Static spin structure factor

Next, we demonstrate the existence of a quantum spin spiral. We first note that such a spiral cannot be detected from the polarization alone since the ground state is translationally invariant (up to numerical errors). In the sector $M_{tot} = S_{tot}$, the expectations $\left\langle \mathbf{S}_i^A\right\rangle$ and $\left\langle \mathbf{S}_i^B\right\rangle$ take values which do not depend on the cell index $i$; for $M_{tot} = 0$, we have $\left\langle \mathbf{S}_i^A\right\rangle = \left\langle \mathbf{S}_i^B\right\rangle = 0$. Thus, we compute the static spin structure factor, and we perform the calculation both in the finite and infinite system.[4] The results are shown in Figs. 8 and 9.

Figure 8 shows the apex-apex structure factor $C^{AA}[\mathbf{S}]$ for various finite $L$ and $J = -1$, $J' = 1$. It features a peak at a small non-zero value of the momentum, and convergence

---

[4]We reiterate that the spin structure factor is defined differently in both cases and partially depends on the choice of $M_{tot}$. Details can be found in Sec. 2.3.

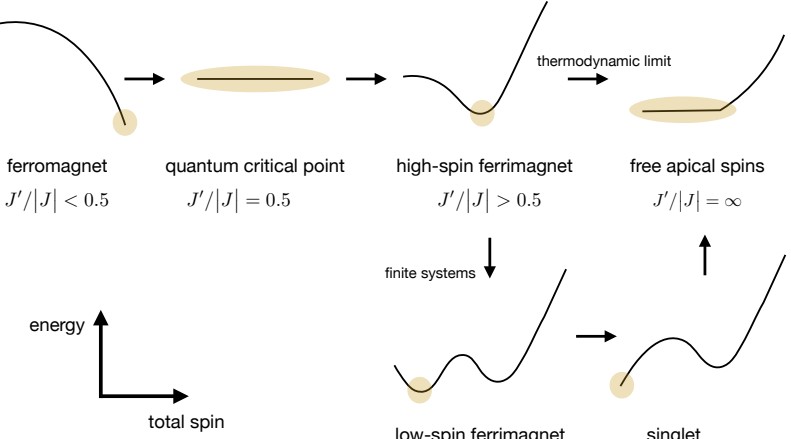

Figure 7: Schematic representation of the energy landscape $E(S_{\text{tot}})$. The parameter $J'/|J|$ is increasing from left to right. The yellow blob indicates the absolute energy minimum. The curves are all exaggerated and the real minima are much more shallow (see Figs. 2 and 5).

For $J'/|J| < 0.5$, the system is a ferromagnet with maximal spin. At the quantum critical point $J'/|J| = 0.5$, we find that $E(S_{\text{tot}})$ is completely flat and that all $S_{\text{tot}}$ values are degenerate for large $L$ within the numerical accuracy (see Fig. 6). For $J'/|J| > 0.5$, we find a ferrimagnet with $0.25 \lesssim S_{\text{tot}}/L \lesssim 0.28$. For finite systems, a low-spin minimum appears if $J'/|J|$ is increased further. Eventually this minimum flattens out and the ground state becomes a singlet. Such a singlet ground state is not found in the thermodynamic limit. Finally, for $J'/|J| = \infty$, the apical spins become free spins and the ground state is degenerate for the values $0 \le S_{\text{tot}}/L \le 1/4$.

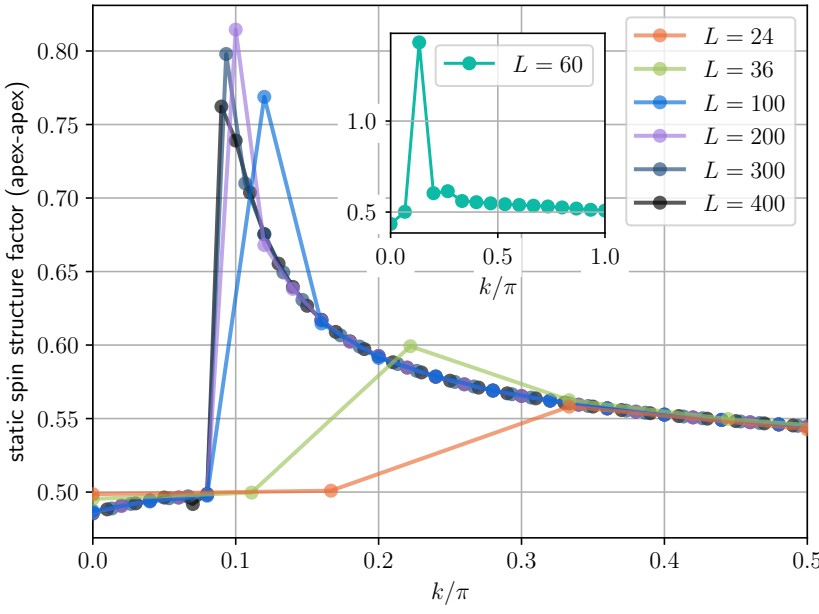

Figure 8: Static apex-apex spin-structure factor as a function of the momentum $k$ for finite systems with periodic boundary conditions and $J = -1$, $J' = 1$ (Eqs. (11) and (12) with $\alpha = \beta = A$).

w.r.t the system size can be reached. For the largest systems of $L = 300, 400$, we find $k_{\text{peak}} = 14/150\pi \approx 0.093\pi$ and $k_{\text{peak}} = 18/200\pi \approx 0.090\pi$. This corresponds to a wavelength of $\lambda_{\text{cells}} = 2\pi/k_{\text{peak}} \approx 21.4 \sim 22.2$ unit cells or $\lambda = 2\lambda_{\text{cells}} \approx 42.9 \sim 44.4$ sites.

While the largest system sizes of $L = 200, 300, 400$ all yield very similar results, there are strong outliers in the smaller systems: For $L = 60$ (inset), we find $k_{\text{peak}} = 4/30\pi \approx 0.13\pi$ or $\lambda = 30$ sites, so that slightly decreasing the wavelength to a value that is commensurate with the system size seems to be energetically favourable. The peak becomes much sharper.

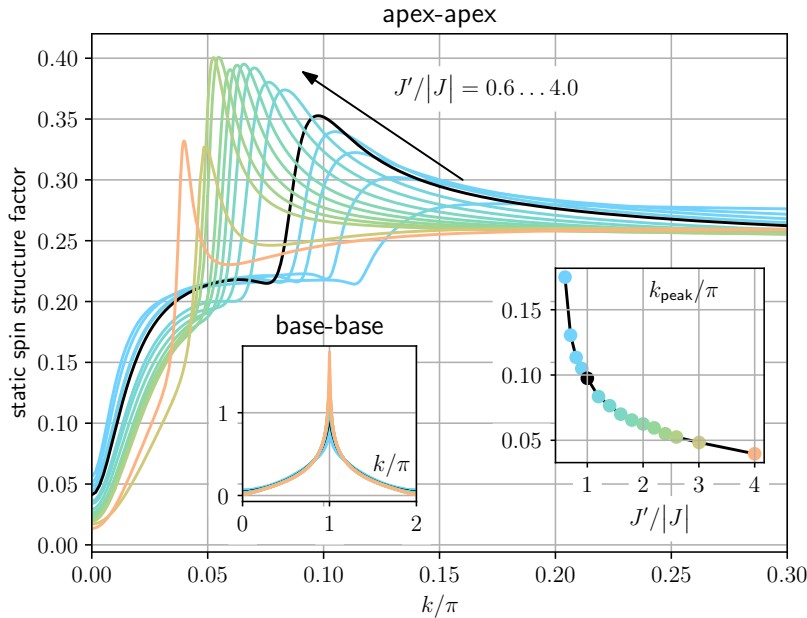

Figure 9: Static apex-apex spin-structure factor for the infinite system with the U(1) spin symmetry exploited (Eqs. (10) and (13) with $\alpha = \beta = A$). The right inset shows the position of the peak as a function of $J'/|J|$ with the same colours as the main plot. The left inset shows the corresponding base-base structure factor ($\alpha = \beta = B$).

Figure 9 shows $C^{AA}[S^z]$ in the infinite system with a continuous $k$ and for different values of $J'/|J| = 1$. For $J'/|J| = 1$, we find $k_{\text{peak}} \approx 0.097\pi$ in agreement with the finite-size calculation. The peak moves closer to zero as $J'/|J|$ is increased. At $J'/|J| = 3$, we have $k_{\text{peak}} \approx 0.048$ and thus roughly a doubled wavelength of about $\lambda \approx 82.8$ sites as compared with $J'/|J| = 1$. We note that the correlations between the basal spins simply remain antiferromagnetic, with a sharp $k_{\text{peak}} = \pi$ (see the left inset of Fig. 9).

In summary, we conclude that the apex spins form a quantum spin spiral with a very long wavelength that increases with $J'/|J|$. Thus, finite-size rings only reflect the behaviour in the thermodynamic limit as long as $L$ accommodates at least several wavelengths. Quantitatively, we find that at least $L \gtrsim 2.5\lambda$ is necessary. This again illustrates that one needs to access large systems and explains the discrepancy with prior results.

In Fig. 10, we study the structure factor for a fixed system size of $L = 100$ as $J'/|J|$ is increased beyond $J'/|J| = 1$. At $J'/|J| = 1.4$, we see that a second peak develops at the smallest possible non-zero $k$-value of $2\pi/50 = 0.04\pi$ ($\lambda = 100$) in addition to the main peak at $6\pi/50 = 0.12\pi$. For $J'/|J| = 1.5$, the main peak shifts to $4\pi/50 = 0.08\pi$ ($\lambda = L/2 = 50$), which coincides with the plateau above $S_{\text{tot}}/L \gtrsim 0.2$ in Fig. 6. Finally, the spiral collapses completely at $J'/|J| = 2$ in favor of pure ferromagnetic alignment with $k_{\text{peak}} = 0$. This means that as soon as the wavelength of the spiral becomes too large, finite-size spirals that form a "standing wave" on the ring with $\lambda = L$ or $\lambda = L/2$ start to compete. Eventually, collinear

alignment becomes energetically favourable and the spiral breaks down. This coincides with the jump to a low-spin state in Figs. 3 and 6, giving more evidence that the low-spin plateau is a finite-size effect: The long-wavelength quantum spin spiral breaks down in a finite system that is too small to host it. In other words, the value of the total spin is related to the wavevector of the spiral; we observe that the long-wavelength spiral is only favorable in combination with a large polarization of $0.25 \lesssim S_{\text{tot}}/L \lesssim 0.28$.

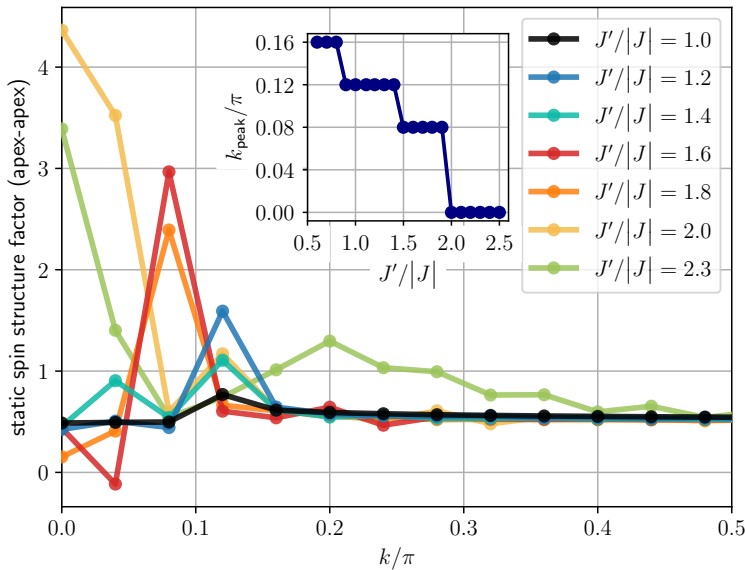

Figure 10: The same as Fig. 8, but for a fixed system size of $L = 100$ and different values of $J'/|J|$. The inset shows the position of the main peak as a function of $J'/|J|$.

# 6 Conclusion

We have demonstrated that the ground state of the FM-AFM sawtooth chain with $J'/|J| > 0.5$ is a ferrimagnet that features an incommensurate quantum spin spiral for the apical spins as well as ordinary antiferromagnetic correlations between the basal spins. The incommensurate behaviour is seen in the spin-spin correlations, while the ground state itself is translationally invariant.

The wavelength of the spiral is large and grows with $J'/|J|$, quickly exceeding sizes $L = 20 - 60$ that are used in typical finite-size calculations (with periodic boundary conditions). By exploiting the SU(2) spin symmetry within our DMRG approach, we are able to accurately treat systems of $L = 200 - 400$ sites with effective bond dimensions in the range of $10^5 - 10^6$. Using the VUMPS formalism, we can tackle the infinite system without finite-size effects at the cost of a lower accuracy. The two methods complement each other and corroborate the above conclusion.

Finally, we have argued that the low-spin plateau found for the FM-AFM sawtooth chain is a finite-size effect related to the competition of the incommensurate infinite-system spiral with finite-system spirals of wavelength $\lambda = L, L/2$. An intriguing question is whether the same physics underlies the Mn wheels (as well as related magnetic molecules [12]), which are finite systems of 70-84 magnetic centres [13–15] and which also exhibit a low-spin ground state stemming from from mixed FM-AFM exchange couplings [15]. In particular, one may wonder whether or not the low-spin state in these systems is also connected to quantum spin

spirals with wavelengths spanning across the whole molecule.

Overall, the FM-AFM sawtooth chain presents an interesting example of the caveats that come with an extrapolation in the system size. Of course, we cannot fully exclude the possibility that additional effects appear on even larger length scales beyond what has been considered here.

An open question for future work is how our observations are affected by an additional apex-apex coupling $\gamma \neq 0$ (see Eq. (6)) or if the case of AFM-AFM couplings also features similar incommensurate behaviour, in particular when polarized by magnetic fields [29].

# Acknowledgements

C.K. acknowledges support by 'Niedersächsisches Vorab' through the 'Quantum- and Nano-Metrology (QUANOMET)' initiative within the project P-1. M.P. is funded by the Deutsche Forschungsgemeinschaft (DFG, German Research Foundation) - project ID 497779765. C.P. is supported by the Deutsche Forschungsgemeinschaft (DFG) through the Cluster of Excellence Advanced Imaging of Matter - EXC 2056 - project ID 390715994. J.S. is funded by the Deutsche Forschungsgemeinschaft (DFG, German Research Foundation) - project ID 449703145 as well as by the Leibniz Supercomputing Center in Garching - project ID pr62to.

# A    Error estimates

## A.1    Energy

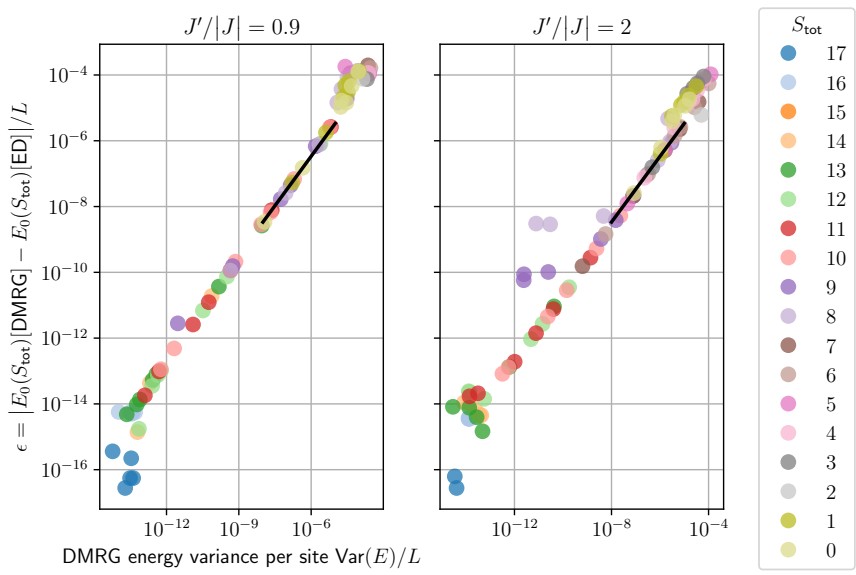

Figure 11: Comparison of the energy density $E_0/L$ computed using DMRG with the exact-diagonalization (ED) result obtained via spinpack [50] for $L = 36$, $J'/|J| = 0.9$ (left) and $J'/|J| = 2$ (right), for various values of the total spin $S_{\text{tot}}$. The DMRG results are computed for different bond dimensions $\chi_{\text{SU(2)}}$, which corresponds to different energy variances per site (see Eq. (7)). The black line is a linear fit, see Eq. (A.14). For $J'/|J| = 2$ and $S_{\text{tot}} = 8, 9$ we were only able to achieve an agreement within $\epsilon \sim 10^{-10} - 10^{-9}$.

In order to translate the energy variance per site (Eq. (7)) into an actual error bar for the energy, we compute the ground state for different energy variances per site and compare with the result of exact diagonalization for $L = 36$ (see Fig. 11).

The typical range of values for the variance that we can achieve for large systems is $\text{Var}(E)/L \sim 10^{-8} - 10^{-5}$. We find that in this regime, the variance is linearly related to the true error $\epsilon$:

$$\epsilon \sim 0.337 \, \text{Var}(E)/L \,. \tag{A.14}$$

The prefactor may of course depend on both $J'/|J|$ and $L$, but Fig. 11 illustrates that it is roughly the same for $J'/|J| = 0.9$ and $J'/|J| = 2$. By comparing with exact-diagonalization results for $L = 16$ (not shown) we find that the relationship still holds.

Thus, we assume that Eq. (A.14) is generally valid, at least as an order-of-magnitude estimate. This allows us to put error bars $\pm \epsilon$ on the energy densities shown in Fig. 2.

### A.2 Finite system: translation invariance

We can check to which extent the ground state of the finite system with periodic boundary conditions is in fact translationally invariant. To this end, we compute $\left\langle \mathbf{S}_i^A \right\rangle$ and $\left\langle \mathbf{S}_i^B \right\rangle$ (for $M_{\text{tot}} = S_{\text{tot}}$) and quantify their spread using the standard deviation of the distribution for all choices $i = 1, 2 \ldots N_{\text{cells}}$. We reiterate that within the SU(2)-symmetric approach, it becomes meaningless to ask for the individual component, one obtains $\left\langle \mathbf{S}_i^{A,B} \right\rangle$ as a scalar number. Figure 12 shows a histogram for $L = 300$, where the results are converged to three digits. Note that we are hereby showing the worst case, and the distribution is even narrower for $L = 100, 200$.

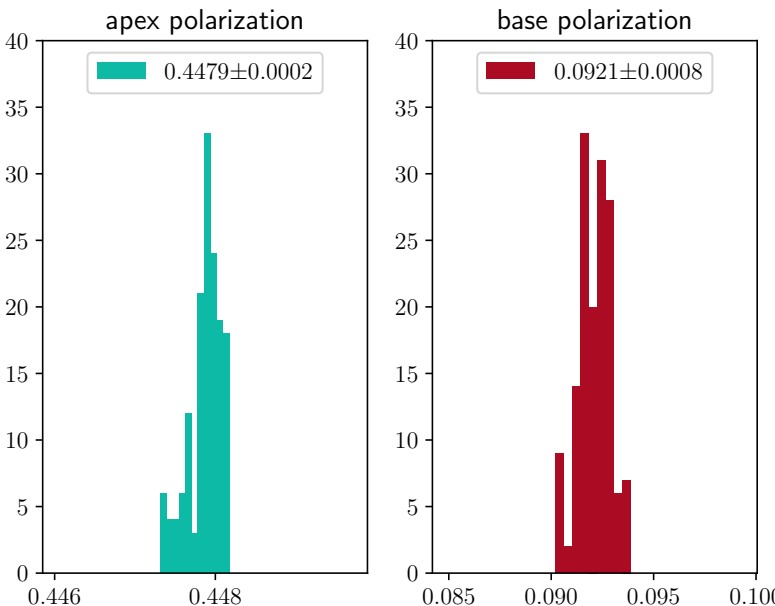

Figure 12: Histogram of the spin polarization $\left\langle \mathbf{S}_i^A \right\rangle$ (apex) and $\left\langle \mathbf{S}_i^B \right\rangle$ (base) for $L = 300$, $S_{\text{tot}}/L = 81/300 = 0.27$, $J = -1$, $J' = 1$ for all values of $i$. The standard deviation of the distribution is taken as the error.

We can repeat the same procedure for a non-local quantity, namely the connected spin-spin correlations of Eq. (9), which should depend only on the distance $d = |l - j|$ and not on the

choice of $j$. We average over all possible choices of $j$:

$$C^{\alpha\beta}[\mathbf{S}](d) := \overline{\left\langle \mathbf{S}_j^\alpha \cdot \mathbf{S}_{j+d}^\beta \right\rangle_c} = \sum_{j=1}^{N_{\text{cells}}} \left\langle \mathbf{S}_j^\alpha \cdot \mathbf{S}_{j+d}^\beta \right\rangle_c, \tag{A.15}$$

and take the standard deviation as a measure of error. Note that this quantity is independent of $M_{\text{tot}}$ due to the SU(2) symmetry. Since the calculation is more costly, we only apply it to selected points. The result for $L = 100, 200, 300$ is displayed in Fig. 13 for $11 \le d \le L/4$ with the corresponding error bars. We see again that the ground state is translationally invariant; the error bars are imperceptible for $L = 100, 200$.

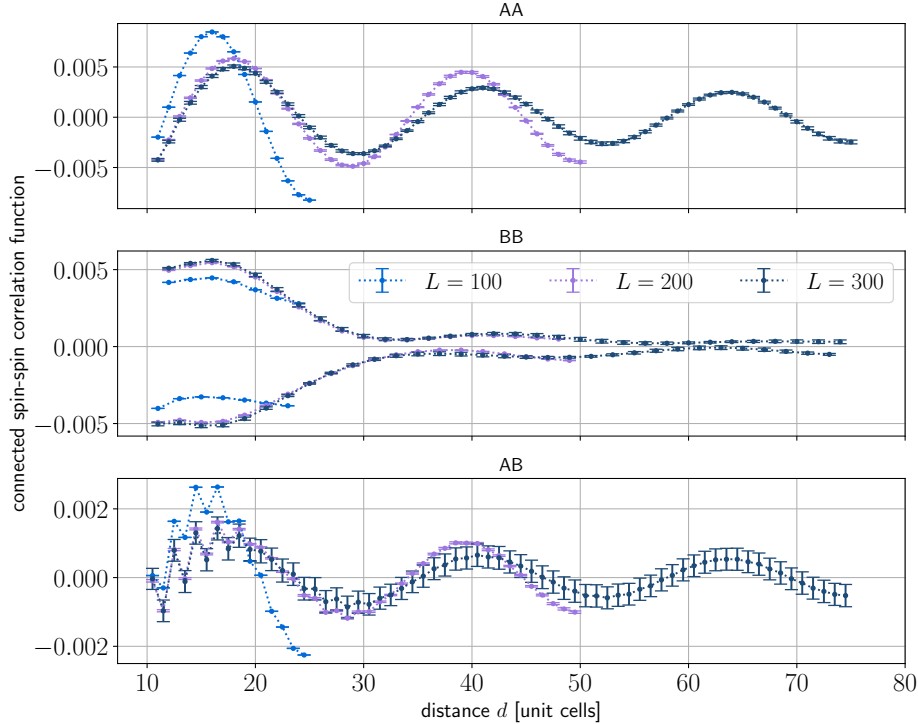

Figure 13: Connected spin-spin correlations (Eq. (A.15)) as a function of the distance measured in unit cells, averaged over all possible initial sites for the apical-apical (AA), basal-basal (BB) and apical-basal (AB) correlations. The standard deviation of the resulting distribution is taken as the error.

## A.3 Infinite systems: error analysis

For the infinite system, we check how the results of Fig. 9 depend on the bond dimension $\chi$. In the simulation, we let $\chi$ grow dynamically and compute the structure factor once the variational error becomes sufficiently small (see Ref. [45] for details). The result for $J = -1$, $J' = 1$ is displayed in Fig. 14. We see that there is no significant change around the main peak, but there is some variation for very small $k$. We extrapolate the result in $\chi^{-1}$ at selected points and find that no appreciable additional peak develops in this region.

## A.4 Comparison between finite and infinite systems

Finally, we compare the full spin-spin correlations $\left\langle \mathbf{S}_j^A \cdot \mathbf{S}_{j+d}^A \right\rangle$ between finite and infinite systems (see Fig. 15). We reiterate that this quantity does not depend on the choice of $M_{\text{tot}}$. The

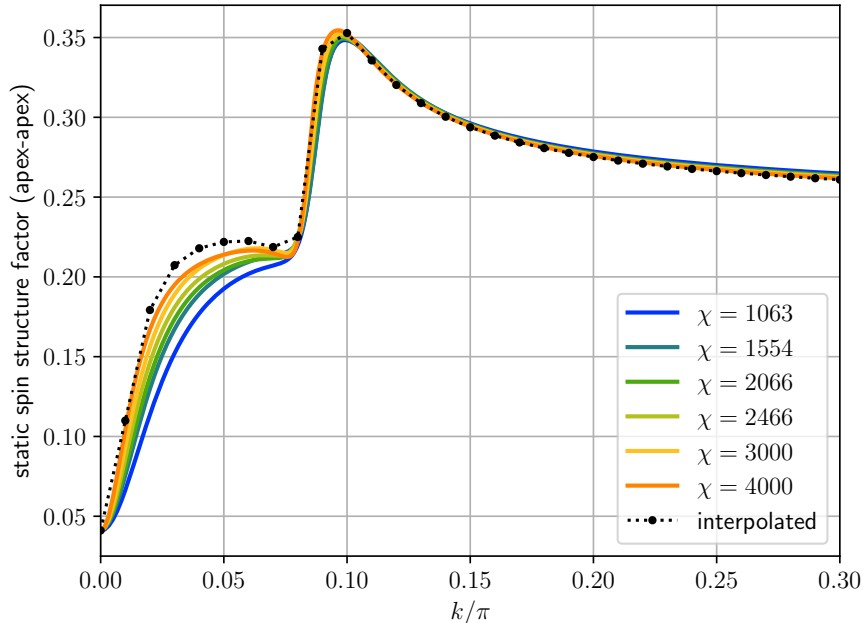

Figure 14: The static spin structure factor of Fig. 9 for $J = -1$, $J' = 1$ and different bond dimensions $\chi$ with the U(1) spin symmetry exploited in the VUMPS algorithm. The result is linearly extrapolated in $\chi^{-1}$ at the selected black points.

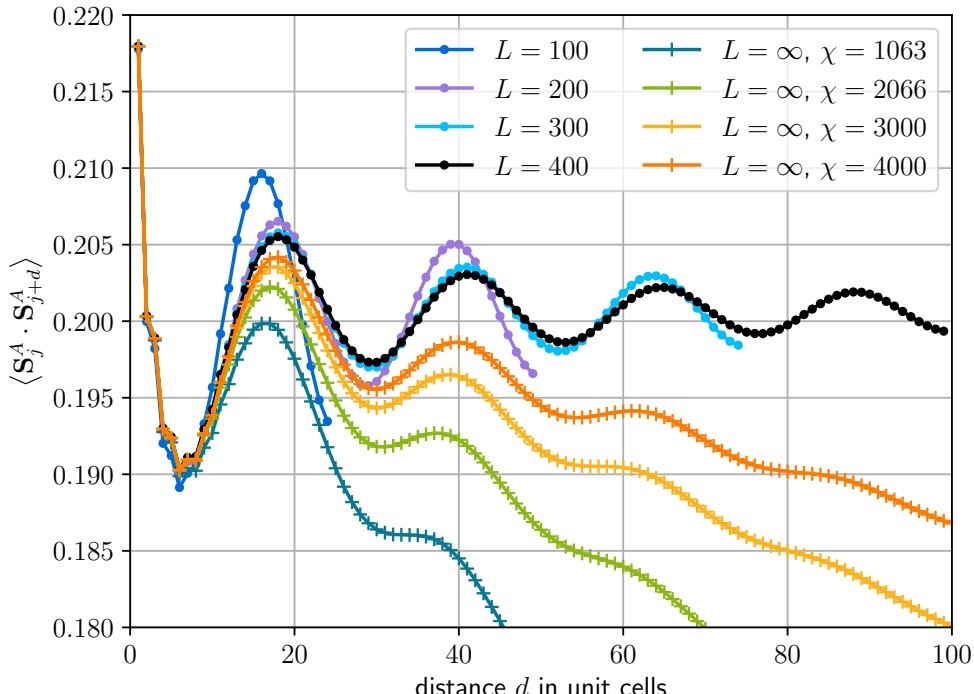

Figure 15: Comparison of the full apex-apex spin correlations for $J = -1$, $J' = 1$ between finite rings of different sizes and the infinite system (with the U(1) symmetry exploited) for various bond dimensions as a function of the distance $d$.

comparison between the curves with $L = 100$ and $L = 400$ indicates that finite-size effects are still manifest for $L = 100$ and $d \geq 10$.

The infinite-system calculation can reproduce the correlations for small distances ($d \leq 10$) rather well even for small bond dimensions $\chi$. In the long-range regime, however, any finite $\chi$ always leads to an exponential decay and thus very large deviations from the $L = 400$ result (which is converged w.r.t. the bond dimension).

Notably, however, the long wavelength of the oscillations is still reproduced. Figure 16 shows that this is almost entirely due to the z-component, so that the quantum spin spiral manifests itself as a peak in the corresponding structure factor (see Sec. 5). This again illustrates that both Eq. (12) and Eq. (13) can be used to demonstrate the existence of the spiral.

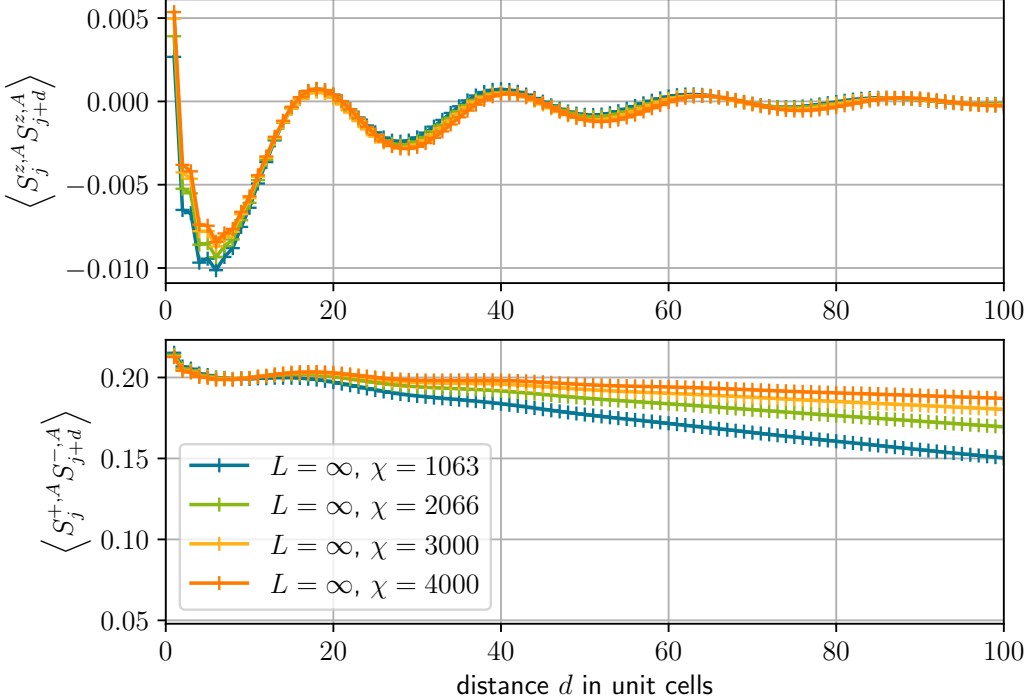

Figure 16: Individual components of the apex-apex spin correlations $\left\langle S_j^{z,A} S_{j+d}^{z,A} \right\rangle$ (top) and $\left\langle S_j^{+,A} S_{j+d}^{-,A} \right\rangle$ (bottom, $S_j^{\pm,\alpha} = S_j^{x,\alpha} \pm i S_j^{y,\alpha}$) computed in the infinite system with U(1) symmetries exploited.

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
