# Peer review of "Quantum spin spiral ground state of the ferrimagnetic sawtooth chain"

_SciPost Physics, doi:SciPost Phys. 14, 052 (2023)_

## Round 1 · Referee Report · Anonymous (Referee 1) · 2022-8-10

Strengths

1- State-of-the-art numerical calculations 2- exhaustive introduction also accessible to non-experts and students, including a review of previous results on the analysed and related models 3- descriptive and well prepared figures 4- exhaustive description of the used numerical methods as well as reasoning on the choice of selected methods and their limitations 5- analyses of the numerical errors 6- clear description of the physical phenomena detected in the model 7- explanation as to why the presented results are different from previous ones on the same model

Report

The authors study a sawtooth chain with FM J interactions between A and B spins and AFM J' interactions on the B chain. The model has already been studied and the existence of a ferromagnetic to ferrimagnetic transition is known. However, the ferrimagnetic phase was believe to be commensurate, while this work claims the ground state is characterised by incommensurate spin spiral correlations between the apical A spins. This results was achieved by exploiting the SU(2) symmetry in the numerical calculations, allowing for a very high value of the effective bond dimension. Moreover, the authors compare their finite-DMRG results with VUMPS calculations for infinite system while also discussing how the latter are limited by the available exploitable symmetries.
They go on to discuss the case with J=-1 and J'=1, which allows them to also underline why their results are different from previous ones thanks to the exploited SU(2) symmetry which allows them to target all possible $S_{tot}$ sectors and compare them to define the energy minimum as well as access fairly large rings. In turn, it also allows them to calculate the correct $S_{tot}/L$ in the ferrimagnetic phase. This shows a different scaling behavior to what previously established. This different behavior was determined thanks to the access to far larger systems, once more reminding the community that large system sizes are often important in determining the physics of strongly correlated systems.
Lastly, the authors discuss the static spin structure factor which allows them to claim incommensurate spin spiral correlations between A spins. Again, they are able to access the incommensurability thanks to large system size. Moreover, they underline how the spiral wavelength is very large for certain J'/|J| values, once more explaining why previous studies had not detected it.

Overall, I believe this work meets the criteria for acceptance in SciPost Physics, in particular it opens a new pathway in an existing research direction, with clear potential for multipronged follow-up work, for example in magnetic molecules such as Mn wheels.
My only comment is to include data and scripts for reproducing the figures presented in the manuscript, especially in the main text.

Requested changes

1- In the general criteria for acceptance in SciPost Physics, it is stated that the paper mus "provide (directly in appendices, or via links to external repositories) all reproducibility-enabling resources: explicit details of experimental protocols, datasets and processing methods, processed data and code snippets used to produce figures, etc.". I urge the authors to provide such repository for reproducing figures, it does not have to contain their DMRG code, but at least the data presented in the main text and, if possible, the script to transform the data to figures.

  • validity: top
  • significance: high
  • originality: high
  • clarity: top
  • formatting: excellent
  • grammar: perfect

Author:  Roman Rausch  on 2022-10-07  [id 2896]

(in reply to Report 1 on 2022-08-10)

We thank the referee for the positive review of our work.

The data and scripts to reproduce the figures are published via the LeoPARD repository of the TU Braunschweig under the link https://leopard.tu-braunschweig.de/receive/dbbs_mods_00071296. A DOI will also be provided.

---

## Round 1 · Referee Report · Anonymous (Referee 2) · 2022-9-1

Strengths

1- state-of-the-art numerical simulations
2- complementary use of finite-size DMRG and infinite-size VUMPS formalism
3- efficient DMRG by exploiting the SU(2) spin symmetry
4- detailed analysis about obtained data
5- possible explanation about differences from previous studies
6- interesting results providing new physics

Weaknesses

1- Only a figure for an optimal case may be shown in Figure 2.

Report

The ferrimagnetic state of the S = 1/2 AFM-AFM sawtooth chain was studied using DMRG technique and VUMPS formalism. Finite-size and infinite-size calculations were performed in a complementary style, and eventually convincing results were provided. Particularly, it is nice idea to directly target the multiplet states by exploiting the SU(2) spin symmetry.

Their findings are very interesting and give a deeper insight in to the field of low-dimensional frustrated magnets: (i) The total spin takes irrational values 0.25<Stot/L<0.28 and reaches 1/4 only in the limit of J’=infinity. (ii) The ground state is characterized by an incommensurate quantum spin spiral in the apex-apex correlations, and on the other hand, the basal-basal correlation is still commensurate.

The manuscript is well written with containing clear comparisons with previous studies as well as careful analyses of obtained data. Therefore, I think that this manuscript meets the acceptance criteria for publication in SciPost Physics. Nevertheless, I just have some questions for the finishing touch. - Figure 2 looks fine but the finite-size effects seem to accidentally small at J=-1 and J’=1. Is it possible to give a similar figure for the other parameter (even without L=252 and L=400 curves) in Appendix. It would be useful to understand the difficulty of studying this system. - It is unclear if a specific Stot is targeted in exact diagonalization calculations including static spin structure factor. Maybe related to this, some values in Figure 3 and 4 look different, for example, Stot/L for L=36 and J’/|J|=1.3. Furthermore, in Figure 4 a sudden change of the Stot/L behaviour for smaller systems between J’/|J|=1.3 and J’/|J|=1.3 is a bit surprising. This seems to be inconsistent with FIG.3 of Ref.5. Do you have any suggestions about it? - f(1/L)=0.25+1/L in Figure 4 should be a straight line because the x-axis is just 1/L. But it seems to be not. What is the reason?

Requested changes

1- Please check the consistency between Figure 3 and 4. 2- If possible, please add a figure like Figure 2 for another parameter in Appendix. 3- Please make it clear whether a specific Stot is targeted in exact diagonalization calculations. 4- Please check the validity of scaling function in Figure 4.

  • validity: high
  • significance: high
  • originality: good
  • clarity: good
  • formatting: excellent
  • grammar: perfect

Author:  Roman Rausch  on 2022-10-07  [id 2897]

(in reply to Report 2 on 2022-09-01)

We thank the referee for the very positive evaluation of our paper.

Our reply to the suggestions is as follows:

>Figure 2 looks fine but the finite-size effects seem to accidentally small at J=-1 and J’=1. Is it possible to give a similar
>figure for the other parameter (even without L=252 and L=400 curves) in Appendix. It would be useful to understand
>the difficulty of studying this system.

We have drawn such a plot (see the new Fig. 5) for the case J'/|J|=1.3 that exhibits the strangely-looking jump in Fig. 3.

>It is unclear if a specific Stot is targeted in exact diagonalization calculations including static spin structure factor.

With ED, we exploit only the Sz symmetry and the conservation of the total momentum. The total spin of the ground state is discerned according to the (2*Stot+1)-fold degeneracies in the spectrum, which we now mention in the main text. The spin structure factor is always computed using the DMRG/VUMPS data. We have added the corresponding remarks to the revised manuscript.

>Maybe related to this, some values in Figure 3 and 4 look different, for example, Stot/L for L=36 and J’/|J|=1.3.

For L=36, the (J'/|J|,Stot) pairs displayed in Fig. 4 are:
0.501 10
0.8 10
1 10
1.3 3
1.5 2
For L=24:
0.501 7
0.8 7
1 7
1.3 1
1.5 1
For L=16:
0.501 5
0.8 0
1 0
1.3 0
1.5 0
This fits with Fig. 3 and we have also checked that ED and DMRG are in agreement here.

>Furthermore, in Figure 4 a sudden change of the Stot/L behaviour for smaller systems between J’/|J|=1.3 and J’/|J|=1.3 is
>a bit surprising. This seems to be inconsistent with FIG.3 of Ref.5. Do you have any suggestions about it?

Indeed, there seems to be a discrepancy for high ratios J’/|J| with Ref. 5. For instance, for J’/|J|=1.5, we find that the high-spin state is never reached for systems of 40-50 sites. For J’/|J|=1.3 the high-spin state is only briefly reached for L=40 and then the system jumps to low spins again.
The only explanation we have is that the energy landscape has a double-minimum structure, whereby the minima may become nearly degenerate (for example around J’/|J|=1.3 and L=40). In this case, one needs to be very careful to land in the right one, which is much easier achieved with SU(2) symmetries and a direct targeting of Stot.
This double-minimum structure can now be seen from the energy landscape plot in the new Fig. 5.

>f(1/L)=0.25+1/L in Figure 4 should be a straight line because the x-axis is just 1/L. But it seems to be not. What is the
>reason?

The reason is simply that we also used a logarithmic scaling on the x-axis. But we realize that this is not really required and have removed it in the revised manuscript.

---

## Round 1 · Referee Report · Anonymous (Referee 3) · 2022-9-7

Strengths

1- Numerics appears to be quite robust 2- Good introduction and explains nicely the problem that is addressed 3- Includes relevant technical details

Weaknesses

1- the advantages of SU(2) in the calculation stated in the abstract and section IIA are manifestly incorrect 2- the author's missed some straightforward followup calculations for some of the symmetry groups (eg the S^2 value for the U(1) symmetric calculations)

Report

The authors study a saw-tooth chain using a combination of finite and infinite MPS methods, using DMRG and VUMPS. This is an interesting model that has a ferrimagnetic state, previously supposed to be commensurate but the authors provide compelling evidence that it is incommensurate.

The use of $SU(2)$ symmetry in the calculations however is misrepresented. Specifically the way the author's calculate the effective bond dimension when using $SU(2)$ is invalid. It is widely known since the start of non-abelian symmetries in DMRG that for non-zero magnetization the advantages of using $SU(2)$ symmetry are lessened. For total spin zero states the correspondence between $SU(2)$ symmetry and $U(1)$ (or no symmetry) is straightforward because there is an exact correspondence that each basis state of spin $\vert j \rangle$ corresponds to a multiplet of $2j+1$ states in a $U(1)$ basis. However no such direct relationship applies to states of non-zero total spin. The author's don't state exactly how they calculate $\chi_\text{eff}$, however they appear to have done something similar to multiplying the maximum number of states kept in the basis by the maximum dimension of the $SU(2)$ representation. This is not a valid way to calculate $\chi_\text{eff}$, as demonstrated by the example of a fully polarized ferromagnet. For a concrete example, suppose the system size is $L=400$, and therefore the fully polarized ferromagnet is solvable with $\chi=1$ $SU(2)$ state, that has total spin $j=200$ at the edge of the system. The author's calculation would determine that this is equivalent to $\chi_\text{eff}=2j+1 = 401$ states using a $U(1)$ basis, however a fully polarized ferromagnetic state of all spins up (for example) is a simple product state that only requires bond dimension $\chi=1$.

I don't know how to to calculate an exact correspondence for $\chi_\text{eff}$ (indeed, there probably isn't an exact correspondence), but to see why the advantages of $SU(2)$ are so restricted for polarized states, one can consider the dimensions of the Hilbert space, visualized on a Bratteli diagram [1]. The dimension of the Hilbert space is the number of ways of getting from one side of the diagram (with spin $s=0$) to the other side (with spin $s=j$). With $U(1)$ symmetry the z-spin can be both positive and negative, so there are many more paths (and hence higher dimensional Hilbert space) than using an $SU(2)$ basis where the spin $s$ is strictly non-negative. However when the magnetization of the state is large, there are clearly very few paths that can venture into negative values of z-spin, and hence the difference in Hilbert space dimension between $U(1)$ and $SU(2)$ basis sets becomes smaller, until finally at full polarization both basis sets have the same dimension.

One way of making a concrete estimate of the advantage of $SU(2)$ symmetry is to compare the relative sizes of the Hilbert spaces, which can be calculated using combinatorial techniques. Eg for L=100 and spin j=27, dimension of the $SU(2)$ Hilbert space in this sector is 17,533,203,421,077,004,122,000, compared with the $U(1)$ sector of $s^z=27$ of 24,865,270,306,254,660,391,200. The ratio is only 1.41818..., so the two basis sets are exploring Hilbert spaces which are very similar in size. Indeed, this is surely an upper bound on the possible value of $\chi_\text{eff}$. So I believe that there is essentially no advantage to using $SU(2)$ symmetry in this calculation. This is also consistent with my own very small scale test - for size $L=100$ magnetization $s=27$ with $\chi=100$ the variational energy using $SU(2)$ and $U(1)$ is barely distinguishable (matching to 8 significant figures). This is also consistent with the author's own results, which indicate that the iMPS results using VUMPS using $U(1)$ symmetry or no symmetry are at least as accurate (if not more so) than the finite size calculations using $SU(2)$, as shown eg in Fig 2.

I am surprised by the author's statement that they don't know of a practical way to calculate the total spin per unit length using $U(1)$ symmetry. MPO techniques to calculate moments of operators, for example $S^2$ (which has a simple MPO form) have been known for a long time [2,3], which is independent of the symmetry group, and it is essentially the same calculation that the author's perform themselves for the correlation function in section IIC using very similar transfer matrix techniques [manuscript ref 47] - the total spin is essentially the integral of the correlation function. But for $U(1)$ a calculation isn't necessary as it can be calculated analytically. The total spin in the thermodynamic limit is
$$
\langle S^2 \rangle = \left \langle \sum_{i,j} \left( S^z_i S^z_j + \frac{1}{2} \left( S^+_i S^-_j + S^-_i S^+_j \right) \right) \right\rangle
$$

For an injective iMPS, the connected part of the 2-point correlation function is necessarily zero at long distances; the only contribution in the limit of $|j-i| \rightarrow \infty$ is from the disconnected part $\langle S^z_i \rangle \langle S^z_j \rangle$ (and likewise for $S^+$ and $S^-$). Now a state with $U(1)$ symmetry is an eigenstate of $S^z$, call it magnetization $s = \langle S^z_i \rangle$ per site. Since the $U(1)$ symmetry is enforced, the local expectation values $\langle S^+_i \rangle$ and $\langle S^-_i \rangle$ are identically zero, hence the leading contribution to $\langle S^2 \rangle$ is $L^2 s^2$, coming from the local $S^z$ contributions. All other contributions are sub-leading ($O(L)$), hence the total magnetization per site $\frac{\sqrt{\langle S^2 \rangle}}{L}$ is identically equal to $s$, being the $U(1)$ quantum number per site. An interesting question is what is the behavior of the sub-leading term, since this is the variance per site of the (square of the) total spin. There is clearly a contribution equal to $sL$ coming from the purely on-site contributions (i.e. the terms with $i=j$ in the summation above), and if there are no other contributions then it is an eigenstate of $S^2$ with eigenvalue $sL(sL+1)$, but this need not necessarily be the case. The remaining term is essentially the integral of the spin-spin correlation function so this should be possible to check.

It is good that the author's calculate the energy variance in the DMRG calculations. This is a much more reliable estimate of the error than the truncation error. However it is unclear whether they use the variance scaling to extrapolate their observables to the limit of zero variance. Also, it is unclear why the authors didn't also calculate the energy variance for the VUMPS calculations, as this is a widely used procedure [1, 4, manuscript ref 47].

[1] https://en.wikipedia.org/wiki/Bratteli_diagram
[2] https://doi.org/10.48550/arXiv.1008.4667
[3] https://doi.org/10.1103/PhysRevB.100.235140
[4] https://doi.org/10.1103/PhysRevB.97.045125

Requested changes

1- the authors should remove/revise the grossly misleading characterization of the efficiency of the finite-size $SU(2)$ calculations. 2- the authors should consider the other comments, but I understand they will not significantly contribute to the overall conclusions of the paper and likely take time time to implement, so this is not required for publication.

  • validity: high
  • significance: good
  • originality: high
  • clarity: high
  • formatting: excellent
  • grammar: excellent

Author:  Roman Rausch  on 2022-10-07  [id 2898]

(in reply to Report 3 on 2022-09-07)

We thank the referee for the recommendation for publication and for the critical remarks on the calculation details which helped us to improve the paper.

Our reply is as follows:

The use of SU(2) symmetry in the calculations however is misrepresented. Specifically the way the author's calculate the effective bond dimension when using SU(2) is invalid. It is widely known since the start of non-abelian symmetries in DMRG that for non-zero magnetization the advantages of using SU(2) symmetry are lessened. For total spin zero states the correspondence between SU(2) symmetry and U(1) (or no symmetry) is straightforward because there is an exact correspondence that each basis state of spin |j⟩ corresponds to a multiplet of 2j+1 states in a U(1) basis. However no such direct relationship applies to states of non-zero total spin. The author's don't state exactly how they calculate χeff, however they appear to have done something similar to multiplying the maximum number of states kept in the basis by the maximum dimension of the SU(2) representation. This is not a valid way to calculate χeff, as demonstrated by the example of a fully polarized ferromagnet. For a concrete example, suppose the system size is L=400, and therefore the fully polarized ferromagnet is solvable with χ=1 SU(2) state, that has total spin j=200 at the edge of the system. The author's calculation would determine that this is equivalent to χeff=2j+1=401 states using a U(1) basis, however a fully polarized ferromagnetic state of all spins up (for example) is a simple product state that only requires bond dimension χ=1.

In the ferromagnetic example, the counting of 2·Stot+1=L+1 states is not wrong per se - it corresponds to the full Hilbert space, and SU(2)-symmetric DMRG is able to provide the information for each value of the projection Mtot. However, in practice one is only interested in the Mtot=Stot=L/2 sector, where the state becomes a product state. This would not be the case for the Stot=L/2 and Mtot=0 sector, which requires a much larger U(1) bond dimension to get accurately.

However, we agree that one shouldn't take the value as a measure of efficiency and it is misleading to call it "effective bond dimension". But we would say that the main reason for that is that one is typically interested in the Mtot=Stot sector, which has less information than is contained in the full SU(2)-symmetric solution. We have amended the revised manuscript accordingly.

One way of making a concrete estimate of the advantage of SU(2) symmetry is to compare the relative sizes of the Hilbert spaces, which can be calculated using combinatorial techniques. Eg for L=100 and spin j=27, dimension of the SU(2) Hilbert space in this sector is 17,533,203,421,077,004,122,000, compared with the U(1) sector of sz=27 of 24,865,270,306,254,660,391,200. The ratio is only 1.41818..., so the two basis sets are exploring Hilbert spaces which are very similar in size. Indeed, this is surely an upper bound on the possible value of χeff. So I believe that there is essentially no advantage to using SU(2) symmetry in this calculation. This is also consistent with my own very small scale test - for size L=100 magnetization s=27 with χ=100 the variational energy using SU(2) and U(1) is barely distinguishable (matching to 8 significant figures). This is also consistent with the author's own results, which indicate that the iMPS results using VUMPS using U(1) symmetry or no symmetry are at least as accurate (if not more so) than the finite size calculations using SU(2), as shown eg in Fig 2.

We agree that SU(2)-symmetric DMRG and U(1)-symmetric DMRG tend to have comparable efficiencies for large Stot and Mtot=Stot. However, there is clearly still a benefit to using SU(2) for this system, as it allows to exactly project out many unwanted low-lying spin states. U(1) may give a non-integer total spin, i.e. may converge to a superposition of different Stot states with the same Mtot. Furthermore, SU(2) allows us to directly plot the energy landscape E(Stot) and see the local minima. Note that the global minimum has a low total spin for large J’/|J| and large systems, which is more difficult to get with an Mtot value close to 0 (see also the reply to report #2). Finally, Fig. 14 clearly shows that U(1) VUMPS and Mtot=0 has trouble reproducing the correlation functions at large distances at comparable bond dimensions, even though the energy matches well enough.

I am surprised by the author's statement that they don't know of a practical way to calculate the total spin per unit length using U(1) symmetry. MPO techniques to calculate moments of operators, for example S^2 (which has a simple MPO form) have been known for a long time [2,3], which is independent of the symmetry group, and it is essentially the same calculation that the author's perform themselves for the correlation function in section IIC using very similar transfer matrix techniques [manuscript ref 47] - the total spin is essentially the integral of the correlation function. But for U(1) a calculation isn't necessary as it can be calculated analytically. The total spin in the thermodynamic limit is ⟨S^2⟩=⟨∑i,j (Szi·Szj+1/2(S+i·S−j+·S−i·S+j))⟩ For an injective iMPS, the connected part of the 2-point correlation function is necessarily zero at long distances; the only contribution in the limit of |j−i|→∞ is from the disconnected part ⟨Szi⟩⟨Szj⟩ (and likewise for S+ and S−). Now a state with U(1) symmetry is an eigenstate of Sz, call it magnetization s=⟨Szi⟩ per site. Since the U(1) symmetry is enforced, the local expectation values ⟨S+i⟩ and ⟨S−i⟩ are identically zero, hence the leading contribution to ⟨S^2⟩ is L^2s^2, coming from the local Sz contributions. All other contributions are sub-leading (O(L)), hence the total magnetization per site √⟨S^2⟩L is identically equal to s, being the U(1) quantum number per site. An interesting question is what is the behavior of the sub-leading term, since this is the variance per site of the (square of the) total spin. There is clearly a contribution equal to s·L coming from the purely on-site contributions (i.e. the terms with i=j in the summation above), and if there are no other contributions then it is an eigenstate of S^2 with eigenvalue sL(sL+1), but this need not necessarily be the case. The remaining term is essentially the integral of the spin-spin correlation function so this should be possible to check.

It is clear how to calculate Stot with U(1) for a finite system using the above formula. For the infinite system, the referee essentially derives our Eq. (8) from the manuscript and it boils down to Stot/L=Mtot/L=⟨S^z_i⟩ (simplifying here to a homogeneous chain). This works whenever Mtot=Stot, which can be achieved by switching off the symmetries altogether, as we mention in the manuscript. This is also the case that is treated in Ref. [3] cited by the referee. We have added Ref. [3] to the list of our references.

However, this approach doesn't work anymore for the sector Mtot=0, which implies that ⟨S^z_i⟩=0. In this case, on the left-hand side of the formula we have ⟨S^2⟩, which scales as L^2. On the right-hand side we have only the connected correlations ∑i,j⟨S^z_i·S^z_j⟩=L∑j⟨S^z_{i0}·S^z_j⟩. It follows that the correlations ∑j(S^z_{i0}·S^z_j) have to scale as L, i.e. reach a constant for long distances, which is not possible with an injective MPS. Hence, we would even conclude that a state with Stot>0, Mtot=0 is not representable as an MPS. In Ref. [3], the second cumulant of the order parameter per site (which is mentioned in the cited paper) is equivalent to the spin structure factor at k=0 and to ⟨S^2⟩/L. Eq. (28) of the paper reads ⟨S^2⟩=k_1^2·L^2+k_2·L. In our case, k_1=0 and k_2 must diverge as L to get a finite total spin. Since the bond dimension sets an effective length scale, it should diverge with increasing bond dimension, corresponding to an increase in the correlation length for ∑j(S^z_{i0}·S^z_j). But what we need is the intensive quantity ⟨S^2⟩/L^2 - it may be possible to find an appropriate scaling, but this is an extrapolation procedure. In Ref. [3], the procedure is instead to compute the non-diverging k_2 in the Mtot=Stot sector.

In the case where Mtot is commensurate with the unit cell, one can set Mtot>0 correspondingly. However, in our case Mtot seems to be irrational and would require an infinite unit cell to get correctly. Hence, this approach is also out of question.

It is good that the author's calculate the energy variance in the DMRG calculations. This is a much more reliable estimate of the error than the truncation error. However it is unclear whether they use the variance scaling to extrapolate their observables to the limit of zero variance.

We have not tried to extrapolate in the variance, but have instead increased the bond dimension to get variances of the order of O(10^{-6}) around the minima; and no larger than O(10^{-5}) everywhere else, as mentioned in App. A1. We have now moved this sentence to the main text.

Also, it is unclear why the authors didn't also calculate the energy variance for the VUMPS calculations, as this is a widely used procedure [1, 4, manuscript ref 47].

Our code uses the two-site variance as described in the original VUMPS paper (Ref. [45] of the manuscript). Error analysis for the infinite system is somewhat confounded by the presence of a variational error (which measures the closeness to the variational minimum) as well as the two-site variance. We typically find that the two-site variance quickly saturates, while the variational error continues to go down. Therefore, our strategy is to make many iterations at each bond dimension in order to achieve a small variational error (ensuring that the result is as well-converged at a given bond dimension as possible) before resizing and then to extrapolate in the inverse bond dimension. We have checked that an extrapolation in the two-site variance yields comparable results. Note that this only concerns the inset of Fig. 2. We now mention the order of magnitude of the two-site variance in the caption.

---

## Round 2 · Referee Report · Anonymous (Referee 1) · 2022-10-11

Report

The authors have added a DOI to their data and made them accesible as required by SciPost. I believe the manuscript should be accepted for publication for the reasons given in my previous report.

---

## Round 2 · Referee Report · Anonymous (Referee 2) · 2022-11-1

Report

The manuscript has been sufficiently improved by addressing all of the referee's suggestions/requests. Especially, new Fig. 5 on the local minima issue and the added data repository URL are very informative for numerical researchers. As I wrote in my previous report, the findings in this paper are very interesting and give a deeper insight in to the field of low-dimensional frustrated magnets. The numerical results seem to be also reliable. Therefore, I recommend a publication of this paper in SciPost Physics.

---

## Round 2 · List of Changes

• data repository URL added
  • new Fig. 5 added to demonstrate the local minima
  • Fig. 4 switched from semilogarithmic to linear
  • descriptions of the numerical procedures and tolerances are revised/rearranged
  • statements on the large "effective bond dimension" are withdrawn according to criticism from report #3
  • new Ref. [48] added

---

## Editorial Decision

published